# Improving Evolutionary Strategies with Generative Neural Networks

## Abstract

Evolutionary Strategies (ES) are a popular family of black-box zeroth-order optimization algorithms which rely on search distributions to efficiently optimize a large variety of objective functions. This paper investigates the potential benefits of using highly flexible search distributions in ES algorithms, in contrast to standard ones (typically Gaussians). We model such distributions with Generative Neural Networks (GNNs) and introduce a new ES algorithm that leverages their expressiveness to accelerate the stochastic search. Because it acts as a plug-in, our approach allows to augment virtually any standard ES algorithm with flexible search distributions. We demonstrate the empirical advantages of this method on a diversity of objective functions.

## 1 Introduction

We are interested in the global minimization of a black-box objective function, only accessible through a *zeroth-order* oracle. In many instances of this problem the objective is expensive to evaluate, which excludes brute force methods as a reasonable mean of optimization. Also, as the objective is potentially non-convex and multi-modal, its global optimization cannot be done greedily but requires a careful balance between exploitation and exploration of the *optimization landscape* (the surface defined by the objective).

The family of algorithms used to tackle such a problem is usually dictated by the *cost* of one evaluation of the objective function (or equivalently, by the maximum number of function evaluations that are reasonable to make) and by a precision requirement. For instance, Bayesian Optimization (Jones et al., 1998; Shahriari et al., 2016) targets problems of very high evaluation cost, where the global minimum must be approximately discovered after a few hundreds of function evaluations. When aiming for a higher precision and hence having a larger budget (e.g. thousands of function evaluations), a popular algorithm class is the one of Evolutionary Strategies (ES) (Rechenberg, 1978; Schwefel, 1977), a family of heuristic search procedures.

ES algorithms rely on a *search distribution*, which role is to propose queries of potentially small value of the objective function. This search distribution is almost always chosen to be a multivariate Gaussian. It is namely the case of the Covariance Matrix Adaptation Evolution Strategies (CMA-ES) (Hansen & Ostermeier, 2001), a state-of-the-art ES algorithm made popular in the machine learning community by its good results on hyper-parameter tuning (Friedrichs & Igel, 2005; Loshchilov & Hutter, 2016). It is also the case for Natural Evolution Strategies (NES) (Wierstra et al., 2008) algorithms, which were recently used for direct policy search in Reinforcement Learning (RL) and shown to compete with state-of-the-art MDP-based RL techniques (Salimans et al., 2017). Occasionally, other distributions have been used; e.g. fat-tails distributions like the Cauchy were shown to outperform the Gaussian for highly multi-modal objectives (Schaul et al., 2011).

We argue in this paper that in ES algorithms, the choice of a standard parametric search distribution (Gaussian, Cauchy, ..) constitutes a *potentially harmful implicit constraint* for the stochastic search of a global minimum. To overcome the limitations of classical parametric search distributions, we propose using *flexible* distributions generated by bijective Generative Neural Networks (GNNs), with computable and differentiable log-probabilities. We discuss why common existing optimization methods in ES algorithms cannot be directly used to train such models and design a tailored algorithm that efficiently train GNNs for an ES objective. We show how this new algorithm can readily incorporate existing ES algorithms that operates on simple search distributions,

---

**Algorithm 1:** Generic ES procedure

---

**input:** zeroth-order oracle on $f$, distribution $\pi_0$, population size $\lambda$
**repeat**

  *(Sampling)* Sample $x_1, \ldots, x_\lambda \overset{\text{i.i.d}}{\sim} \pi_t$
  *(Evaluation)* Evaluate $f(x_1), \ldots, f(x_n)$.
  *(Update)* Update $\pi_t$ to produce $x$ of potentially smaller objective values.
**until** convergence;

---

like the Gaussian. On a variety of objective functions, we show that this extension can significantly accelerate ES algorithms.

We formally introduce the problem and provide background on Evolutionary Strategies in Section 2. We discuss the role of GNNs in generating flexible search distributions in Section 3. We explain why usual algorithms fail to train GNNs for an ES objective and introduce a new algorithm in Section 4. Finally we report experimental results in Section 5.

## 2 PRELIMINARIES

In what follows, the real-valued objective function $f$ is defined over a compact $\mathcal{X}$ and $\pi$ will generically denote a probability density function over $\mathcal{X}$. We consider the global optimization of $f$:

$$x^* \in \underset{x \in \mathcal{X}}{\operatorname{argmin}} f(x) \tag{1}$$

### 2.1 EVOLUTIONARY STRATEGIES

The generic procedure followed by ES algorithms is presented in Algorithm 1. To make the update step tractable, the search distribution is tied to a family of distributions and parametrized by a real-valued parameter vector $\theta$ (e.g. the mean and covariance matrix of a Gaussian), and is referred to as $\pi_\theta$. This update step constitutes the main difference between ES algorithms.

**Natural Evolution Strategies**   One principled way to perform that update is to minimize the expected objective value over samples $x$ drawn from $\pi_\theta$. Indeed, when the search distribution is parametric and tied to a parameter $\theta$, this objective can be differentiated with respect to $\theta$ thanks to the log-trick:

$$J(\theta) \triangleq \mathbb{E}_{\pi_\theta}\left[f(x)\right] \qquad \text{and} \qquad \frac{\partial J(\theta)}{\partial \theta} = \mathbb{E}_{\pi_\theta}\left[f(x)\frac{\partial \log \pi_\theta(x)}{\partial \theta}\right] \tag{2}$$

This quantity can be approximated from samples - it is known as the score-function or REINFORCE (Williams, 1992) estimator, and provides a direction of update for $\theta$. Unfortunately, naively following a stochastic version of the gradient (2) – a procedure called Plain Gradient Evolutionary Strategies (PGES) – is known to be highly ineffective. PGES main limitation resides in its instability when the search distribution is concentrating, making it unable to *precisely* locate any local minimum. To improve over the PGES algorithm the authors of Wierstra et al. (2008) proposed to descend $J(\theta)$ along its *natural gradient* (Amari, 1998). More precisely, they introduce a trust-region optimization scheme to limit the instability of PGES, and minimize a linear approximation of $J(\theta)$ under a Kullback-Leibler (KL) divergence constraint:

$$\underset{\delta\theta}{\operatorname{argmin}} \quad J(\theta + \delta\theta) \simeq J(\theta) + \delta\theta^T \nabla_\theta J(\theta) \quad \text{s.t} \quad \operatorname{KL}(\pi_{\theta+\delta\theta}||\pi_\theta) \leq \epsilon \tag{3}$$

To avoid solving analytically the trust region problem (3), Wierstra et al. (2008) shows that its solution can be approximated by:

$$\delta\theta^* \propto -F_\theta^{-1} \nabla_\theta J(\theta) \quad \text{where} \quad F_\theta = \mathbb{E}_{\pi_\theta}\left[\nabla_\theta \log \pi_\theta(x) \nabla_\theta \log \pi_\theta(x)^T\right] \tag{4}$$

is the Fischer Information Matrix (FIM) of $\pi_\theta$. The parameter $\theta$ is therefore not updated along the negative gradient of $J$ but rather along $F_\theta^{-1} \nabla_\theta J(\theta)$, a quantity known as the natural gradient. The FIM $F_\theta$ is known analytically when $\pi_\theta$ is a multivariate Gaussian and the resulting algorithm, Exponential Natural Evolutionary Strategies (xNES) (Glasmachers et al., 2010) has been shown to reach state-of-the-art performances on a large ES benchmark.

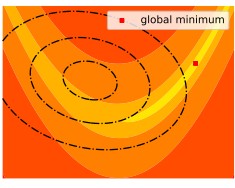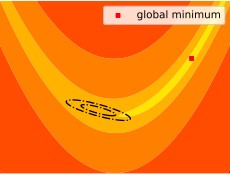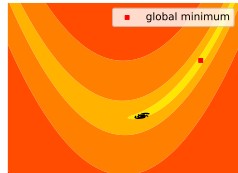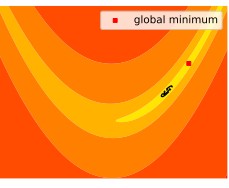

*ES iterations*

Figure 2: Example of an undesirable behavior of a Gaussian search distribution. The dashed lines represent density level lines of the search distribution. Because the latter cannot have a curved profile, it is forced to drastically reduce its entropy until it reaches the straight part of the valley.

**CMA-ES**   Naturally, there exist other strategies to update the search distribution $\pi_\theta$. For instance, CMA-ES relies on a variety of heuristic mechanisms like covariance matrix adaptation and evolution paths, but is only defined when $\pi_\theta$ is a multivariate Gaussian. Explaining such mechanisms would be out of the scope of this paper, but the interested reader is referred to the work of Hansen (2016) for a detailed tutorial on CMA-ES.

## 2.2   Limitations of classical search distributions

ES implicitly balance the need for exploration and exploitation of the optimization landscape. The exploitation phase consists in updating the search distribution, and exploration happens when samples are drawn from the search distribution's tails. The key role of the search distribution is therefore to produce a support adapted to the landscape's structure, so that new points are likely to improve over previous samples.

We argue here that the choice of a given parametric distribution (the multivariate Gaussian distribution being overwhelmingly represented in state-of-the-art ES algorithms) constitutes a *potentially harmful implicit constraint* for the stochastic search of a global minimum. For instance, a Gaussian distribution is not adapted to navigate a curved valley because of its inability to continuously curve its density. This lack of flexibility will lead it to drastically reduce its entropy, until the curved valley looks *locally* straight. At this point, the ES algorithm resembles a hill-climber and barely takes advantage of the exploration abilities of the search distribution. An illustration of this phenomenon is presented in Figure 2 on the Rosenbrock function. Another limitation of classical search distribution is their inability to follow *multiple hypothesis*, that is to explore at the same time different local minima. Even if mixture models can show such flexibility, hyper-parameters like the number of mixtures have optimal values that are impossible to guess *a priori*.

We want to introduce *flexible* search distributions to overcome these limitations. Such distributions should, despite their expressiveness, be easily trainable. We should also be concerned when designing them with their role in the exploration/exploitation trade off: a search distribution with too much capacity could over-fit some seemingly good samples, leading to premature convergence. To sum-up, we want to design search-distributions that are:

- more flexible than classical distributions
- yet easily trainable
- while keeping control over the exploration / exploitation trade-off

In the following section, we carefully investigate the class of Generative Neural Networks (GNNs) to find a parametric class of distributions satisfying such properties.

## 3   Flexible search distributions with GNNs

Generative Neural Networks (MacKay, 1995) have been studied in the context of *density estimation* and shown to be able to model complex and highly multimodal distributions (Srivastava et al., 2017). We propose here to leverage their expressiveness for ES, and train them in a principled way thanks to the ES objective:

$$J(\pi) = \mathbb{E}_\pi \left[ f(x) \right]$$

As discussed in Section 2, optimizing $J(\pi)$ with gradient-based methods is possible through the score-function estimator, which requires to be able to compute and efficiently differentiate the log-probabilities of $\pi$.

## 3.1 GNN BACKGROUND

The core idea behind a GNN is to map a *latent* variable $z \in \mathcal{Z}$ drawn from a known distribution $\nu_\omega$ to an output variable $x = g_\eta(z)$ where $g_\eta$ is the forward-pass of a neural network. The parameter $\eta$ represents the weights of this neural network while $\omega$ describe the degrees of freedom of the latent space distribution $\nu_\omega$. We denote $\theta = (\omega, \eta)$ and $\pi_\theta(x)$ the density of the output variable $x$.

For general neural network architectures, it is impossible to compute $\pi_\theta(x)$ for samples $x$ drawn from the GNN. This is namely why their are often trained with adversarial methods (Goodfellow et al., 2014) for sample generation purposes, bypassing the need of computing densities, but at the expense of a good density estimation (mode-dropping). An alternative to adversarial methods was proposed with variational auto-encoders (Kingma & Welling, 2013) however at the cost of learning two neural networks (an encoder and a decoder). A less computationally expensive method consists in restricting the possible architectures to build *bijective* GNNs, also known as Normalizing Flows (NF) (Rezende & Mohamed, 2015; Papamakarios et al., 2017), which allows the exact computation of the distribution's density. Indeed, if $g_\eta$ is a bijection from $\mathcal{Z}$ to $\mathcal{X}$ with inverse $h_\eta \triangleq g_\eta^{-1}$, the change of variable formula provides a way to compute $\pi_\theta(x)$:

$$\pi_\theta(x) = \nu_\omega(h_\eta(x)) \cdot \left| \frac{\partial h_\eta(x)}{\partial x} \right| \tag{5}$$

To have a tractable density one therefore needs to ensure that the determinant of the Jacobian $|\partial h_\eta(x)/\partial x|$ is easily computable. Several models satisfying these two properties (*i.e* bijectivity and computable Jacobian) have been proposed for density estimation (Rippel & Adams, 2013; Dinh et al., 2014; 2016), and proved their expressiveness despite their relatively simple structure.

NFs therefore answer two of our needs when building our new search distribution: flexibility and easiness to train. In this work, we will focus on one NF model: the Non-Linear Independent Component Estimation (Dinh et al., 2014) (NICE) model, for its numerical stability and *volume preserving* properties.

## 3.2 NICE MODEL

The authors of NICE proposed to build complex yet invertible transformations through the use of *additive coupling layers*. An additive coupling layer leaves half of its input unchanged, and adds a non-linear transformation of the first half to the second half. More formally, by noting $v = [v_1, v_2]$ the output of a coupling layer and $u = [u_1, u_2]$ its input, one has:

$$v_1 = u_1 \quad \text{and} \quad v_2 = u_2 + t(u_1) \tag{6}$$

where $t$ is an arbitrarily complex transformation - modelled by a Multi-Layer Perceptron (MLP) with learnable weights and biases. This transformation has unit Jacobian determinant and is easily invertible:

$$u_1 = v_1 \quad \text{and} \quad u_2 = v_2 - t(v_1) \tag{7}$$

and only requires a feed-forward pass on the MLP $t$. The choice of the decomposition $u = [u_1, u_2]$ can be arbitrary, and is performed by applying a binary filter to the input. By stacking additive coupling layers, one can create complex distributions, and the inversion of the resulting mapping is independent of the complexity of the neural networks $t$. The density of the resulting distribution is readily computable thanks to the inverse transform theorem (5).

## 3.3 VOLUME PRESERVING PROPERTIES

The transformation induced by NICE is *volume preserving* (it has a unitary Jacobian determinant). This is quite desirable in a ES context, as the role of concentrating the distribution on a minimum can be left to the latent space distribution $\nu_\omega$. The role of the additive coupling layers is therefore only to introduce non-linearities in the inverse transform $h_\eta$ so that the distribution is better adapted

to the optimization landscape. The fact that this fit is volume-preserving (every subset of the latent space has an image in the data space with the same probability mass) encourages the search distribution to align its tails with regions of small value of the optimization landscape, which is likely to improve the quality of future exploration steps. The NICE model therefore fits perfectly our needs for a flexible search distribution that is easy to train, and that provides enough control on the exploration / exploitation trade-off. Other bijective GNN models like the Real-NVP (Dinh et al., 2016) introduce non-volume preserving transformations, which cannot provide such a control. In practice, we observed that using such transformations for ES led to early concentration and premature convergence.

## 4 AN EFFICIENT TRAINING ALGORITHM

We are now equipped with enough tools to use GNNs for ES: an adapted model (NICE) for our search distribution $\pi_\theta$, and an objective to train it with:

$$J(\theta) = \mathbb{E}_{\pi_\theta}[f(x)] \tag{8}$$

Here, $\theta$ describes *jointly* the free parameters of the latent distribution $\nu_\omega$ and $\eta$, the weights and biases of the MLPs forming the additive coupling layers.

We start this section by explaining why existing training strategies based on the objective (8) are not sufficient to truly leverage the flexibility of GNNs for ES, before introducing a new algorithm tailored for this task.

### 4.1 LIMITATIONS OF EXISTING TRAINING STRATEGIES

We found that the PGES algorithm (naive stochastic gradient descent of (8) with the score-function estimator) applied to the NICE distribution suffers from the same limitations as when applied to the Gaussian; it is unable to precisely locate any local minimum. As for the Gaussian, training the NICE distribution for ES requires employing more sophisticated algorithms - such as NES.

However, using the natural gradient for the GNNs distributions is not trivial. First the Fischer Information Matrix $F_\theta$ is not known analytically and must be estimated via Monte-Carlo sampling, thereby introducing approximation errors. Also, we found that the approximations justifying to follow the descent direction provided by the natural gradient are not adapted to the NICE distribution. Indeed, the assumption behind the NES update (4) is that the loss $J(\theta)$ can be (locally) well approximated by the quadratic objective:

$$J(\theta + \delta\theta) = J(\theta) + \delta\theta^T \nabla_\theta J(\theta) + \frac{\gamma}{2}\delta\theta^T F_\theta \delta\theta \tag{9}$$

where $\gamma$ is a given non-negative Lagrange multiplier. For NICE, given the highly non-linear nature of $\pi_\theta$ this approximation is bound to fail even close to the current parameter $\theta$ and will lead to spurious updates. A classical technique (Martens, 2010) to avoid such updates is to artificially increase the curvature of the quadratic term, and is known as *damping*. Practically, this implies using $F_\theta + \beta I$ instead of $F_\theta$ as the local curvature metric, with $\beta$ a non-negative damping parameter.

We found that to ensure continuous decrease of $J(\theta)$, and because of its highly non-linear nature when using the GNNs, the damping parameter $\beta$ has to be set to such high values that the modifications of the search distribution are too small to quickly make progress and by no means reaches state-of-the-art performances. We observed that even if the training of the additive coupling layers is performed correctly (i.e the distribution has the correct *shape*), high damping of the latent space parameters prevents the distribution from quickly concentrating when a minimum is found.

It is unclear how the damping parameter should be adapted to avoid spurious update, while still allowing the distribution to make large step in the latent space and ensure fast concentration when needed. In the following, we present an *alternated minimization* scheme to bypass the issues raised by natural gradient training for GNN distributions in a ES context.

### 4.2 ALTERNATING MINIMIZATION

So far, we used the parameter $\theta$ to describe both $\omega$ and $\eta$ (respectively, the free parameters of the latent space distribution $\nu_\omega$ and the degrees of freedom of the non-linear mapping $g_\eta$), and the

optimization over all these parameters was performed jointly. Separating the roles of $\omega$ and $\eta$, the initial objective (2) can be rewritten as follows:

$$J(\theta) = \mathbb{E}_{z \sim \nu_\omega} \left[ f(g_\eta(z)) \right] = J(\omega, \eta) \tag{10}$$

Therefore, the initial objective can be rewritten as the expected value of samples drawn from the latent distribution, under the objective $f \circ g_\eta$ - that is, the representation of the objective function $f$ in the latent space. If $\nu_\omega$ is a standard distribution (i.e efficiently trainable with the natural gradient) and $f \circ g_\eta$ is a *well structured* function (i.e one for which $\nu_\omega$ is an efficient search distribution), then the single optimization of $\omega$ by classical methods (such as the natural gradient) should avoid the limitations discussed in 2.2. This new representation motivates the design of a new training algorithm that optimizes the parameters $\omega$ and $\eta$ *separately*.

**Alternating Minimization**   In the following, we will replace the notation $\pi_\theta$ with $\pi_{\omega,\eta}$ to refer to the NICE distribution with parameter $\theta = (\omega, \eta)$. We want to optimize $\omega$ and $\eta$ in an alternate fashion, which means performing the following updates at every step of the ES procedure:

$$\omega_{t+1} = \underset{\omega}{\operatorname{argmin}} \, J(\omega, \eta_t) \tag{11a}$$

$$\eta_{t+1} = \underset{\eta}{\operatorname{argmin}} \, J(\omega_{t+1}, \eta) \tag{11b}$$

This means that at iteration $t$, samples are drawn from $\pi_{\omega_t, \eta_t}$ and serve to first optimize the latent space distribution parameters $\omega$, and then the additive coupling layers parameters $\eta$. For the following iteration, the population is sampled under $\pi_{\omega_{t+1}, \eta_{t+1}}$.

The update (11a) of the latent space parameters is naturally derived from the new representation (10) of the initial objective. Indeed, $\omega$ can be updated via natural gradient ascent of $J(\omega, \eta_t)$ - that is with keeping $\eta = \eta_t$ fixed. Practically, this therefore reduces to applying a NES algorithm to the latent distribution $\nu_\omega$ on the modified objective function $f \circ g_{\eta_t}$.

Once the latent space parameters updated, the coupling layers parameters should be optimized with respect to:

$$J(\omega_{t+1}, \eta) = \mathbb{E}_{\pi_{\omega_{t+1}, \eta}} \left[ f(x) \right] \tag{12}$$

At this stage, the only available samples are drawn under $\pi_{\omega_t, \eta_t}$. To estimate, based on these samples, expectations under $\pi_{\omega_{t+1}, \eta_t}$ one must use *importance propensity scores*:

$$J(\omega_{t+1}, \eta) = \mathbb{E}_{\pi_{\omega_t, \eta_t}} \left[ f(x) \frac{\pi_{\omega_{t+1}, \eta}(x)}{\pi_{\omega_t, \eta_t}(x)} \right] \tag{13}$$

The straightforward minimization of this *off-line* objective is known to lead to degeneracies (Swaminathan & Joachims, 2015, Section 4), and must therefore be regularized. For our application, it is also desirable to make sure that the update $\eta$ does not undo the progress made in the latent space - in other words, we want to regularize the change in $f \circ g_\eta$. To that extent, we adopt a technique proposed in Schulman et al. (2017) and minimize a modification on the initial objective with *clipped* propensity weights:

$$\eta_{t+1} = \underset{\eta}{\operatorname{argmin}} \quad \mathbb{E}_{\pi_{\omega_{t+1}, \eta_t}} \left[ f(x) \operatorname{clip}_\varepsilon \left( \frac{\pi_{\omega_{t+1}, \eta}(x)}{\pi_{\omega_{t+1}, \eta_t}(x)} \right) \right] \tag{14}$$

$\operatorname{clip}_\varepsilon(x)$ clips the value of $x$ between $1 - \epsilon$ and $1 + \epsilon$. The parameter $\varepsilon$ is an hyper-parameter that controls the change in distribution, and the program (14) can be efficiently solved via a gradient descent type algorithm, such as Adam (Kingma & Ba, 2014).

To sum up, we propose optimizing the latent distribution and the coupling layers separately. The latent space is optimized by natural gradient descent, and the coupling layers via an off-policy objective with clipped propensity weights. We call this algorithm GNN-ES for Generative Neural Networks Evolutionary Strategies.

**Latent space optimization**   It turns out the GNN-ES can be readily modified to incorporate virtually *any* existing ES algorithms that operates on the simple distribution $\nu_\omega$. For instance, if $\nu_\omega$ is set to be a multivariate Gaussian with learnable mean and covariance matrix, the latent space optimization (11a) can be performed by either xNES or CMA-ES. This holds for any standard distribution $\nu_\omega$ and any ES algorithm operating on that distribution. This remark allows us to place GNN-ES in a more general framework and to understand it as a way to improve existing ES algorithm, by providing a principled way to learn complex, non-linear transformations on top of rather standard search distributions (like the Gaussian). In what follows, we will use the GNN prefix in front of existing ES algorithm to describe its augmented version with our algorithm, working as a *plug-in*. Pseudo-code for this general algorithm can be found in Appendix B.

### 4.3   ADDITIONAL TOOLS

**Using historic data**   ES algorithms typically use small *populations* of samples to estimate expectations. Such small sample sizes don't allow for enough data exposure for the GNN to build a meaningful transformation $g_\eta$. To circumvent this problem, we augment the off-line program (14) with samples for past generations thanks to the *fused importance sampling* estimator (Peshkin & Shelton, 2002). This technique is classical in similar settings like MDP-based reinforcement learning and counterfactual reasoning (Nedelec et al., 2017; Agarwal et al., 2017) and proves to be essential for our problem. Formally, for a given horizon $T$ that controls how far we look in the past, this amounts to storing the samples $x$ drawn from $\pi_{\theta_{t-T+1}}, \ldots, \pi_{\theta_t}$ (as well as their respective scores) in a buffer $\mathcal{H}_T$. The objective (13) can then be rewritten as:

$$\mathbb{E}_{\pi_{\omega_t,\eta_t}} \left[ f(x) \frac{\pi_{\omega_{t+1},\eta}(x)}{\pi_{\omega_t,\eta_t}(x)} \right] = T \cdot \mathbb{E}_{x,f(x)\in\mathcal{H}_T} \left[ f(x) \frac{\pi_{\omega_{t+1},\eta}(x)}{\pi_{\theta_{t-T+1}}(x) + \ldots + \pi_{\theta_t}(x)} \right] \qquad (15)$$

This technique allows to increase the data exposure of the GNN by using past samples (and therefore does not require additional function evaluations) and to reduce the variance of the off-line estimator of the original expectation (12) (Nedelec et al., 2017). To control the change in distribution, the fused propensity weights can then be clipped in a similar fashion than in the program (14).

**Mode preserving properties**   To achieve improved exploration, the search distribution should align its tails with the level sets of the objective function. This is not guaranteed when performing the update step (14) since the GNN's update could simply move the mean of the search distribution without shaping the tails. One way to encourage the GNN's capacity to be allocated to the tails is to impose a *mode-preserving* property. If $\mu$ denotes the location of a mode of the latent distribution, then the mode of the distribution $\pi_\theta$ generated by the NICE model is located in $g_\eta(\mu)$ (see Appendix A for the proof). It is therefore easy to build a map $f_\eta$ based on the initial $g_\eta$ that is mode-preserving:

$$f_\eta(z) \triangleq g_\eta(z) - g_\eta(\mu) + f_{\eta_t}(\mu) \qquad (16)$$

where $\mu_t$ denotes the mode of the latent distribution $\nu_\omega$ at iteration $t$. Defined as such, $f_\eta$ preserves the mode of the previous search distribution (since $f_{\eta_{t+1}}(\mu) = f_{\eta_t}(\mu)$), is trivially still a bijection and remains volume preserving. Using the push-forward map $f_\eta$ instead of $g_\eta$, we explicitly push the flexibility brought by the GNN to impact only the tails of the search distribution. As detailed in an ablation study presented in Appendix F, this additional tool turns out to be essential in order to use GNNs for ES.

## 5   EXPERIMENTAL RESULTS

In all that follows, we build the NICE model with three coupling layers. Each coupling layer's non-linear mapping $t$ is built with a one hidden layer MLP, with 128 neurons and leaky ReLU (Maas et al., 2013) activation functions. This architecture is kept constant in all our experiments.

### 5.1   VISUALIZATION

We present here two-dimensional visualizations of the behavior of a GNN distribution trained with GNN-xNES - the latent distribution is therefore Gaussian. Figure 3a displays the density level lines

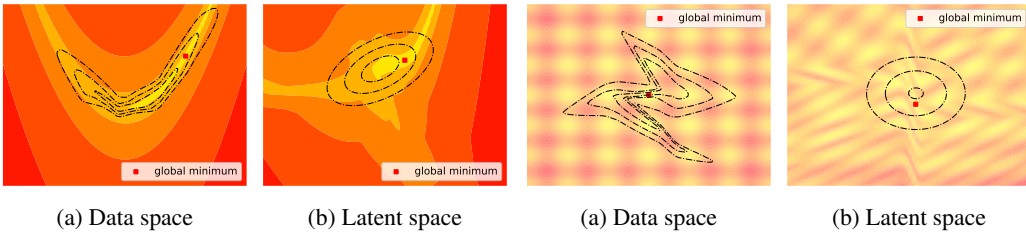

| (a) Data space | (b) Latent space | (a) Data space | (b) Latent space |

Figure 3: Rosenbrock        Figure 4: Rastrigin

Density level curves (dotted lines) in the data space and in the latent space

of the resulting search distribution on the Rosenbrock function. Figure 3b displays the density level lines of the latent distribution, as well as the learned representation of the objective in the latent space. The search distribution is able to have curved density isolines, enabling better exploration. In the latent space, the global minimum can be reached without navigating a curved valley. Figures 4a and 4b provide similar visualizations on the Rastrigin function, a highly multimodal but symmetric objective. The GNN lowers the barriers between local minima, making it easier to escape a local minimum to the global minimum.

## 5.2 Synthetic objectives

**Experimental set-up** We present experiments on both unimodal and multimodal objectives for xNES and GNN-xNES. We use the official implementation of xNES[1] with default hyper-parameters (such as the population size $\lambda$), both as a baseline and as an inner optimization method for GNN-xNES. All experiments are run on the COmparing COntinous Optimizers (COCO) (Hansen et al., 2016) platform, a popular framework for comparing black-box optimization algorithms. It namely allows to benchmark different algorithms on translated and rotated versions of the same objectives, in order to evaluate multiple configurations with different global minimum positions. We compare xNES and GNN-xNES on functions from the 2018 Black-Box Optimization Benchmark (BBOB) (Hansen et al., 2010) suite. When comparing these two algorithms, we impose that their initial search distributions are close in order to ensure fair comparison. We insist on the fact that the xNES algorithm has the exact same configuration whether it is used by itself or as an inner-optimization algorithm for GNN-xNES. Further experimental details, including additional hyper-parameters value for GNN-xNES are provided in Appendix C.

**Unimodal landscapes** We run the different algorithms on two unimodal landscapes where we expect GNN search distributions to bring a significant improvement over the Gaussian - as discussed in 2.2. These objectives functions are the Rotated Rosenbrock function (a curved valley with high conditioning) and the Bent Cigar (an asymmetric and curved Cigar function). Extensive details on these objective functions can be found in the BBOB documentation (Hansen et al., 2010). Results on additional unimodal functions can be found in Appendix E.

Performance is measured through Empirical Cumulative Distribution Functions (ECDFs) of the runtime, also known as data profiles (Moré & Wild, 2009). Such curves report the fraction of problems solved as a function of the number of objective evaluations. For a given precision $\Delta$, a problem is said to be solved if the best function evaluation made so far is smaller than $f(x^*) + \Delta$. We create 200 problems, equally spaced on a log-scale from $\Delta = 10^2$ to $\Delta = 10^{-5}$ and, as in the COCO framework, aggregate them over 15 function instances. Results are presented in Figure 5 for the two benchmark functions and in dimensions $d = 2, 5, 10$.

**Multimodal landscapes** We now compare the performances of the different algorithms on a collection of three multimodal objectives: the Rastrigin function, the Griewank-Rosenbrock function and the Schwefel function. Extensive details about these objectives can be found in Hansen et al. (2010).

---

[1]available in the PyBrain library (Schaul et al., 2010)

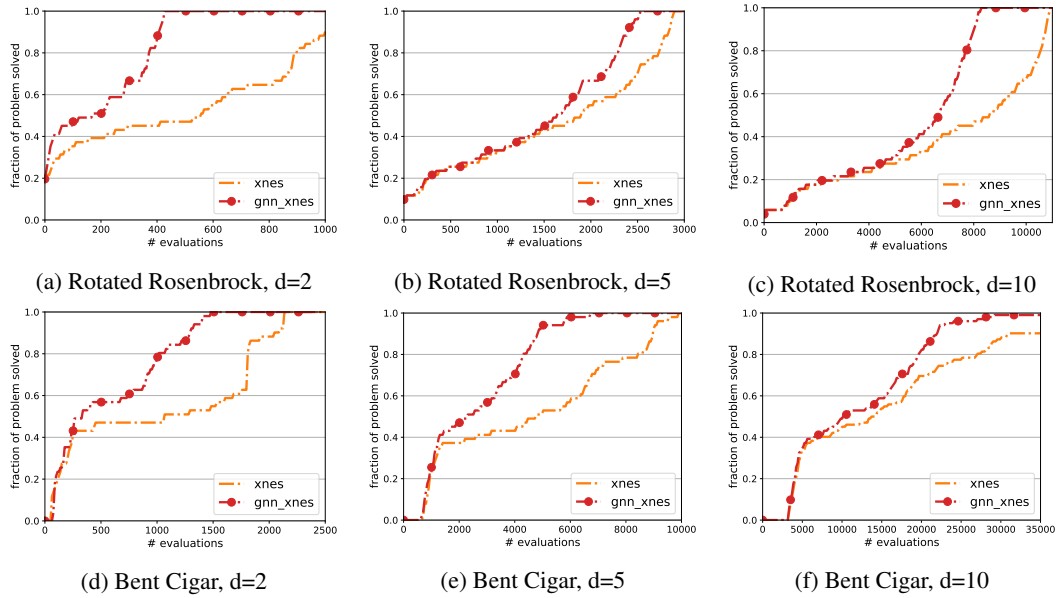

Figure 5: ECDFs curves comparing GNN-xNES and xNES on the Rotated Rosenbrock and Bent Cigar functions, in dimensions d=2,5,10.

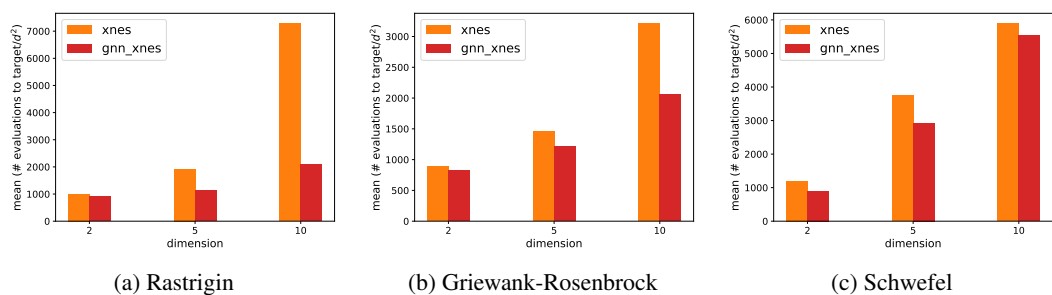

Figure 6: Scaling comparison of GNN-xNES and xNES on the Rastrigin, Griewank-Rosenbrock and Schwefel functions, d=2,5,10.

When using ES algorithms to optimize multimodal functions, it is usual to augment them with *restart strategies* (Hansen, 2016). When convergence is detected, the search distribution is re-initialized in order to search another part of the landscape, and often the population size is increased. This allows to fairly compared algorithms that converge fast to potentially bad local minima, and algorithms that converges slower to better minima. Their exist a large variety of restart strategies (Loshchilov et al., 2012; Auger & Hansen, 2005); as the official implementation of xNES is not equipped with a default one, we trigger a restart whenever the algorithm makes no progress for more than $30 \times d$ iterations. The standard deviation of the search distribution is set back to 1, and its mean sampled uniformly within the compact $\mathcal{X}$ of interest (defined by the COCO framework). At each restart, the population size of the algorithm is multiplied by 2, as in Auger & Hansen (2005). This restart strategy is used for both xNES and GNN-xNES.

We measure performance as the number of functions evaluations to find an objective value smaller than $f(x^*) + 10^{-5}$ within a budget of $d \times 10^5$ function evaluations, averaged over 15 function instances. When an algorithm is not able to discover the global minimum within the given budget, we use the maximum number of evaluations as its performance. For visualization purposes, this measure of performance is divided by $d^2$. Results are reported in Figure 6. On all objectives, and for all dimensions, GNN-xNES discovers (in average) the global minimum faster than xNES. Additional results on others multimodal functions are presented in Appendix E.

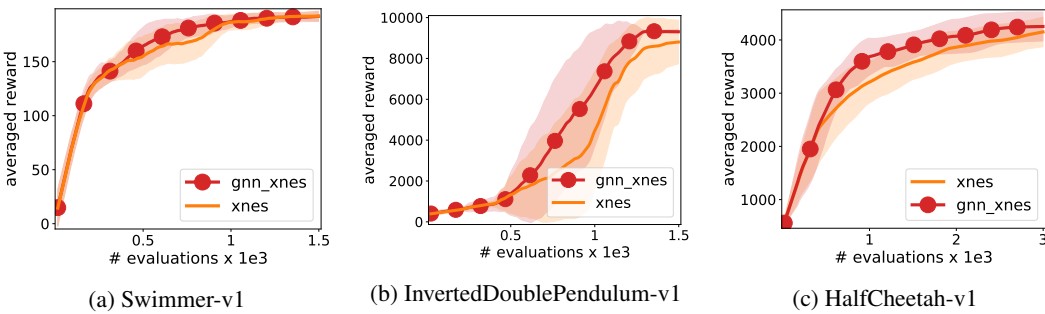

| (a) Swimmer-v1 | (b) InvertedDoublePendulum-v1 | (c) HalfCheetah-v1 |

Figure 7: Direct Policy Search experiments

## 5.3 REINFORCEMENT LEARNING EXPERIMENTS

The goal of this section is to present additional comparison between xNES and GNN-xNES on RL-based objective functions - less synthetic than the previously considered BBOB functions. ES algorithms have recently been used for direct policy search in Reinforcement Learning (RL) and shown to reach performances comparable with state-of-the-art MDP-based techniques (Liu et al., 2019; Salimans et al., 2017). Direct Policy Search ignores the MDP structure of the RL environment and rather considers it as a *black-box*. The search for the optimal policy is performed directly in parameter space to maximize the average reward per trajectory:

$$f(x) = \mathbb{E}_{\tau \sim p_x} \left[ \sum_{j \in \tau} r_j \right] \tag{17}$$

where $p_x$ is the distribution of trajectories induced by the policy (the state-conditional distribution over actions) parametrized by $x$, and $r$ the rewards generated by the environment. The objective (17) can readily be approximated from samples by simply rolling out $M$ trajectories, and optimized using ES. In our experiments[2], we set $M = 10$ and optimize deterministic linear policies (as in Rajeswaran et al. (2017)).

In Figures 7a and 7b we report results of the GNN-xNES algorithm compared to xNES, when run on the Mujoco locomotion tasks Swimmer and InvertedDoublePendulum, both from the OpenAI Gym (Brockman et al., 2016). Performance is measured by the average reward per trajectory as a function of the number of evaluations of the objective $f$. Results are averaged over 5 random seeds (ruling the initialization of the environment and the initial distribution over the policy parameters $x$). In all three environments, GNN-xNES discovers behaviors of high rewards faster than xNES.

## 6 CONCLUSION

In this work, we motivate the use of GNNs for improving Evolutionary Strategies by pinpointing the limitations of classical search distributions, commonly used by standard ES algorithms. We propose a new algorithm that leverages the high flexibility of distributions generated by bijective GNNs with an ES objective. We highlight that this algorithm can be seen as a plug-in extension to existing ES algorithms, and therefore can virtually incorporate *any* of them. Finally, we show its empirical advantages across a diversity of synthetic objective functions, as well as from objectives coming from Reinforcement Learning. Beyond the proposal of this algorithm, we believe that our work highlights the role of *expressiveness* in exploration for optimization tasks. This idea could be leverage in other settings where exploration is crucial, such a MDP-based policy search methods. An interesting line of future work could focus on optimizing GNN-based conditional distribution for RL tasks - an idea already developed in Ward et al. (2019); Mazoure et al. (2019). Other possible extensions to our work could focus on investigating first-order and mixed oracles, such as in Grathwohl et al. (2017); Faury et al. (2018).

---

[2]We used the `rllab` library (Duan et al., 2016) for our experiments.

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

## A    COMPUTING THE MODE OF THE SEARCH DISTRIBUTION

We prove here the fact that if $\mu$ denotes the location of the mode of the latent distribution $\nu_\omega$, then $g_\eta(\mu)$ is a mode for $\pi_{\omega,\eta}$. Indeed, under reasonable smoothness assumptions, one has that $y$ is a mode for $\pi_{\omega,\eta}$ if and only if:

$$\left. \frac{\partial \pi_{\omega,\eta}(x)}{\partial x} \right|_{x=y} = 0 \tag{18}$$

Since $\pi_{\omega,\eta}(x) = \nu_\omega(h_\eta(x))$, this is therefore equivalent to:

$$\left. \frac{\partial h_\eta(x)}{\partial x} \right|_{x=y} \cdot \left. \frac{\partial \nu_\omega(z)}{\partial z} \right|_{z=h_\eta(y)} = 0 \tag{19}$$

In the NICE model, we have that $\left| \frac{\partial h_\eta(x)}{\partial x} \right| = 1$ for all $x$ hence the matrix $\left. \frac{\partial h_\eta(x)}{\partial x} \right|_{x=y}$ is invertible and its kernel is reduced to the null vector. Therefore:

$$\left. \frac{\partial \nu_\omega(z)}{\partial z} \right|_{z=h_\eta(y)} = 0 \tag{20}$$

and therefore $\mu = h_\eta(y)$ by definition of $\mu$ (the only critical point of $\nu_\omega$). Hence since $h_\eta^{-1} = g_\eta$, we have that $y = g_\eta(\mu)$ which concludes the proof.

## B    ALGORITHM PSEUDO-CODE

We provide below the pseudo-code for the generic algorithm GNN-$\mathcal{A}$-ES, where $\mathcal{A}$ is a generic ES algorithm operating on a parametric distribution $\nu_\omega$. The additional hyper-parameters are the horizon $T$ as well as the clipping constant $\varepsilon$. The function clip$(x, lb, ub)$ clips the input $x$ between a lower-bound $lb$ and an upper-bound $ub$.

---

**Algorithm 2:** GNN-$\mathcal{A}$-ES (ex: GNN-xNES, GNN-CMA-ES)

---

**inputs**            : objective function $f$, distribution $\nu_\omega$ and its related ES algorithm $\mathcal{A}$
**hyper-parameters:** clipping constant $\varepsilon$, NICE model architecture, initial parameters $\omega_0$, initial
                    weights $\eta_0$, horizon $T$, population size $\lambda$

*(Initialization)*
    Initialize NICE MLPs weights and biases with $\eta_0$.
    Let $\mathcal{H}$ be a circular buffer of length $T \times \lambda$
**while** *not terminate* **do**
    *(Sampling)*
       Sample $Z = \{z_1, \ldots z_\lambda\} \overset{\text{i.i.d}}{\sim} \nu_{\omega_t}$
       Apply $f_{\eta_t}$ to Z, obtain $X = \{x_1, \ldots x_\lambda\} \overset{\text{i.i.d}}{\sim} \pi_{\omega_t,\eta_t}$
       Evaluate $F = \{f(x_1), \ldots, f(x_\lambda)\}$.
       Let $\mathcal{H} \leftarrow \mathcal{H} + \{X, F, \pi_{\omega_t,\eta_t}\}$
    *(ES update)*
       //One step ES-based optimization of the latent space
       Apply $\mathcal{A}$ to the latent distribution

$$\omega_{t+1} \leftarrow \mathcal{A}\left(\nu_{\omega_t}, (Z, F)\right)$$

    *(GNN update)**
       //Many-steps gradient based optimization of the GNN
    $\eta_{t+1} \simeq \eta\left\{\sum_{x,f \in \mathcal{H}} f \cdot \text{clip}\left(\frac{\pi_{\omega_{t+1},\eta}(x)}{\sum_{\pi \in \mathcal{H}} \pi(x)}, \frac{\pi_{\omega_{t+1},\eta_t}(x)}{\sum_{\pi \in \mathcal{H}} \pi(x)} \cdot (1 - \varepsilon), \frac{\pi_{\omega_{t+1},\eta_t}(x)}{\sum_{\pi \in \mathcal{H}} \pi(x)} \cdot (1 + \varepsilon)\right)\right\}$
**end**

---

*The *(GNN iteration)* step can be performed with virtually any gradient descent solver. In all our experiments, we used Adam (Kingma & Ba, 2014) with learning rate 1e-4 for 500 epochs.

Algorithm 2 does not detail the mode-preserving addition for the sake of readability and clarity. We provide additional details on this procedure here. Let $\mu_t$ be the mode of the latent distribution $\nu_{\omega_t}$. At the *(Initialization)* step, set $\alpha_0 = g_{\eta_0}(\mu_0)$ where $g_\eta(\cdot)$ is the push-forward map on the NICE model described in Section 3.2. For all round $t \geq 1$, let $f_\eta(z) = g_\eta(z) - g_\eta(\mu_t) + \alpha_t$. The variable $\alpha_t$ represent the push forward mapping of the latent distribution's mean under the current model. Every time the latent space is updated - the *(ES update)* step, let $\alpha_{t+1} = f_{\eta_t}(\mu_{t+1})$. Then, for the *(GNN update)*, optimize the forward-map $f_\eta(z) = g_\eta(z) - g_\eta(\mu_{t+1}) + \alpha_{t+1}$. After this update, we have $f_{\eta_{t+1}}(\mu_{t+1}) = \alpha_{t+1} = f_{\eta_t}(\mu_{t+1})$, which means that the mode of the search distribution (which is the image of the latent distribution mode) has not been impacted by the GNN update.

## C  EXPERIMENTAL DETAILS

### C.1  HYPER-PARAMETERS

**Baselines**  We use xNES with its default (adapted) hyper-parameters (described in Wierstra et al. (2008)) for both its baselines versions and its inner optimization parts in GNN-xNES. The population size $\lambda$ is one such hyper-parameters, and is therefore set to $\lambda = 4 + \lfloor 3 \log(d) \rfloor$. Also, as it is classically done in ES algorithms, we use a rank-based *fitness shaping*, designed to make the algorithm invariant with respect to order-preserving cost transformations. We use the same fitness-shaping function as in Wierstra et al. (2008).

**GNN-ES**  Across all experiments, we use the same hyper-parameters for GNN-xNES without fine tuning for each tasks. We use three coupling layers, each with a single hidden layer MLP with 128 hidden neurons and Leaky ReLU activations. The MLPs are initialized via Glorot initialization, and the clipping constant is set to $\varepsilon = 0.05$. The history size $T$ was determined experimentally, and set to $T = \lfloor 3 * (1 + \log(d)) \rfloor$. When restarts are used, this history size is divided by the numbers of restart so far (as the population size grows larger).

### C.2  SYNTHETIC OBJECTIVES

Every synthetic objective we used in this work was taken from the BBOB2019 benchmark dataset. Their expression as well as additional details on the framework can be found in Hansen et al. (2010; 2016). At the beginning of each experiment, we set the Gaussian search distribution (for xNES) and the Gaussian latent distribution (for GNN-xNES) to a standard normal, with a mean uniformly sampled within the compact $\mathcal{X}$ of interest (defined by the COCO framework).

### C.3  RL ENVIRONMENTS

Table 1 provides details on the RL environment used to compare GNN-xNES and xNES, like the dimensions of the state space $\mathcal{S}$ and action space $\mathcal{A}$, the number $d$ of the policy's degrees of freedom and the maximum number of steps $m$ per trajectory. At the beginning of each experiment, we set the Gaussian search distribution (for xNES) and the Gaussian latent distribution (for GNN-xNES) to a standard normal with zero mean. In this particular case, where the function evaluations are noisy, we kept the default population size of the xNES algorithm.

| Name | $|\mathcal{S}|$ | $|\mathcal{A}|$ | d | m |
|---|---|---|---|---|
| Swimmer-v1 | 13 | 2 | 28 | 1000 |
| InvertedDoublePendulum-v1 | 11 | 1 | 12 | 1000 |
| HalfCheetah-v1 | 20 | 6 | 126 | 1000 |

Table 1: Reinforcement Learning environments

## D  TWO-DIMENSIONAL VISUALIZATIONS

We provide in Figure 8 additional two-dimensional visualizations of the behavior of GNN-xNES, on the Rosenbrock, Rastrigin, Beale and Bent-Cigar functions. We see that the NICE distributions can

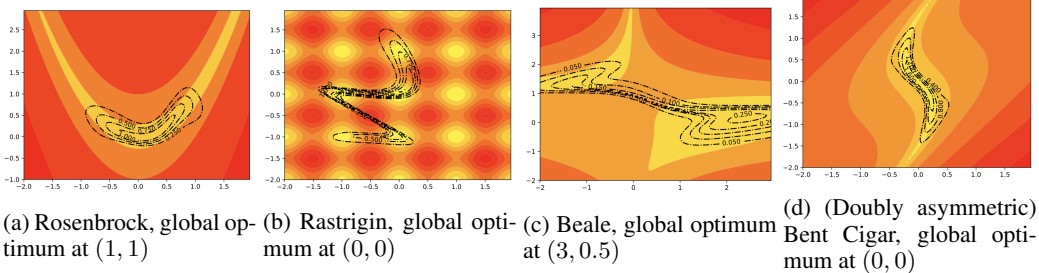

(a) Rosenbrock, global optimum at $(1, 1)$

(b) Rastrigin, global optimum at $(0, 0)$

(c) Beale, global optimum at $(3, 0.5)$

(d) (Doubly asymmetric) Bent Cigar, global optimum at $(0, 0)$

Figure 8: Two-dimensional visualizations. The black dotted lines represent the isolines of the level curves of a NICE search distribution trained with GNN-xNES.

| Algorithm | mean(# restarts), d=2 | mean(# restarts), d=5 | mean(# restarts), d=10 |
|---|---|---|---|
| **xNES** | 2.3 | 2.7 | 3.4 |
| **GNN-xNES** | 1.3 | 2.5 | 2.9 |

Table 2: Mean number of restarts needed to discover the global minimum on the Rastrigin function.

efficiently fit each optimization landscapes, without having to reduce its entropy like a multivariate normal would.

## E ADDITIONAL RESULTS

We present here some additional results on some unimodal and multimodal synthetic functions. Figure 9 present ECDFs curve obtained from the Attractive Sector function, a highly asymmetrical function around its global minimum. On such a function, GNN-xNES seems to accelerate xNES in small dimensions, however this speed-up disappears in higher dimensions. Figure 10 presents results on the Rosenbrock function (without random rotations). Again, GNN-xNES accelerates the xNES algorithm. Figure 11 present results on the multimodal functions Gallagher's Gaussian 101 Peaks and Gallagher's Gaussian 21 Peaks. Again, GNN-xNES discovers the global minimum faster (on average) than xNES.

In our multimodal experiments, we used simulated restarts as a fair mean of comparing different algorithm (this is common practice in order to fairly compare algorithms that converge fast to potentially bad local minima to algorithms that converge slowly to the global minimum). If the empirical results prove that GNN-xNES accelerate xNES in the discovery of the global minimum, it does not prove that GNN-xNES leverages the flexibility of the GNN to detect the global minimum when xNES misses it. In an attempt to prove that it is indeed the case, we report in Table 2 the number of restarts needed by both GNN-xNES and xNES to discover the global minimum on the Rastrigin function (averaged over the 15 randomly initialized run). For this instance, GNN-xNES consistently discovers the global minimum with less restarts than xNES.

As detailed in Section 4, one can apply Algorithm 2 as a plug-in to any ES method. So far, we empirically evaluated the benefits of our approach by comparing xNES against its GNN extension (GNN-xNES). We present in Figure 12 additional evaluations obtained by comparing CMA-ES and its GNN extension (denoted GNN-CMA-ES) on the Rosenbrock function in dimension 2,5 and 10. CMA-ES is considered to be the state-of-the-art ES algorithm, and improving its performances is a non-trivial task. On the considered example GNN-CMA-ES improves CMA-ES, highlighting the empirical benefit of our approach for a large class of ES algorithm. One can however observe that the performance boost brought by the GNN extension is milder for GNN-CMA-ES then for GNN-xNES. We suspect that this is due to the use of *cumulation* via an evolution path in CMA-ES[3], which basically introduces a momentum-like update when optimizing the latent distribution. While using an evolution path makes a lot of sense when optimizing a stationary objective, it can be quite harmful for non-stationary ones. We therefore believe that the cumulation step in CMA-ES (for the latent distribution) and the GNN optimization (making the objective optimized by CMA-ES in the

---

[3]We used the PyCMA library (Hansen et al., 2019) for these experiments.

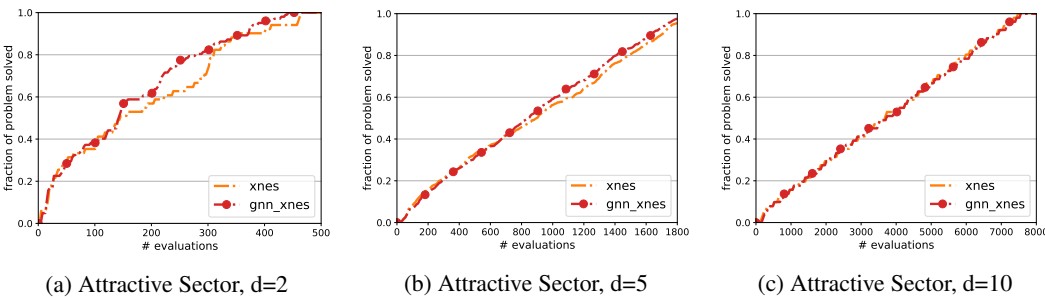

Figure 9: ECDFs curve for the Attractive Sector function, d=2,5,10

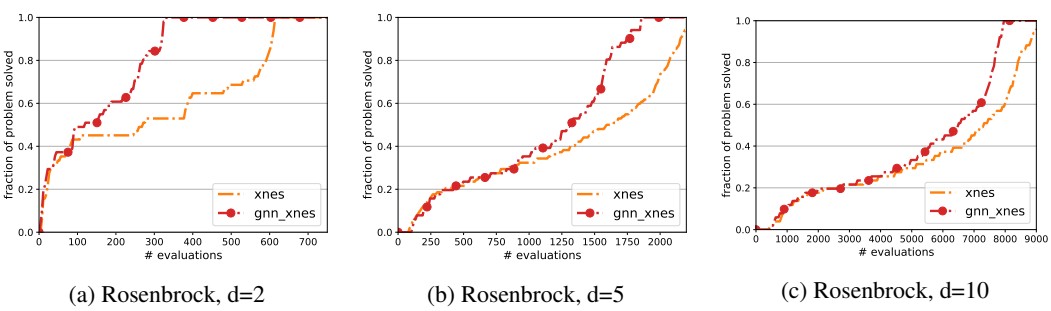

Figure 10: ECDFs curve for the Rosenbrock function, d=2,5,10

latent space non-stationary) can lead to conflicting updates and might hinder the benefits brought by the GNN's additional flexibility. Designing a GNN update strategy complying with the use of evolution paths could therefore be a way of further improving GNN-CMA-ES, and is left for future work.

# F  ABLATION STUDY

We present here an ablation study for two additional tools that we introduced after the alternating optimization view: the mode preserving (16) extension as well as the history augmentation (15). Figure 13 presents ECDFs curves on the Rosenbrock, Rotated Rosenbrock and Bent Cigar functions in 2D, for a version of GNN-xNES that doesn't use history but only the current population. Using history and therefore exposing the GNN to larger datasets improves the procedure. Figure 14 present similar results on a version of GNN-xNES without the mode preserving property (16). Again, one can notice that ensuring that the GNN training is mode-preserving is crucial to improve experimental results.

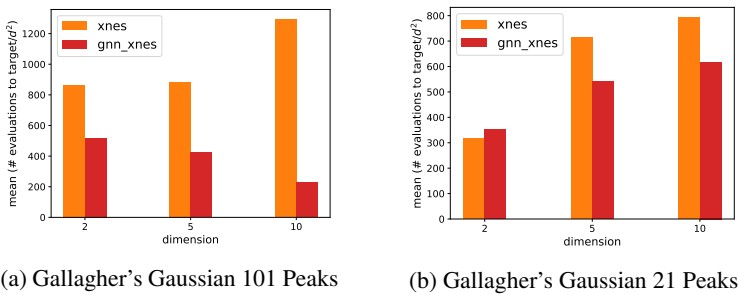

(a) Gallagher's Gaussian 101 Peaks

(b) Gallagher's Gaussian 21 Peaks

Figure 11: Scaling comparison of GNN-xNES and xNES on the Gallagher's Gaussian 101 Peaks and Gallagher's Gaussian 21 Peaks functions, d=2,5,10.

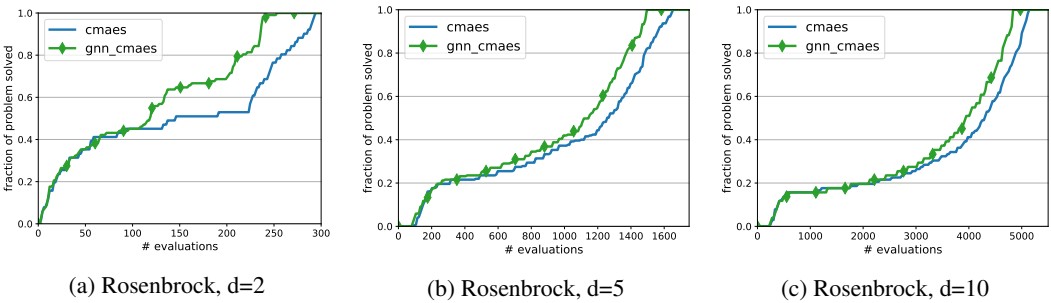

(a) Rosenbrock, d=2          (b) Rosenbrock, d=5          (c) Rosenbrock, d=10

Figure 12: ECDFs curves for the Rosenbrock function, (d=2,5,10) comparing the CMA-ES and GNN-CMA-ES

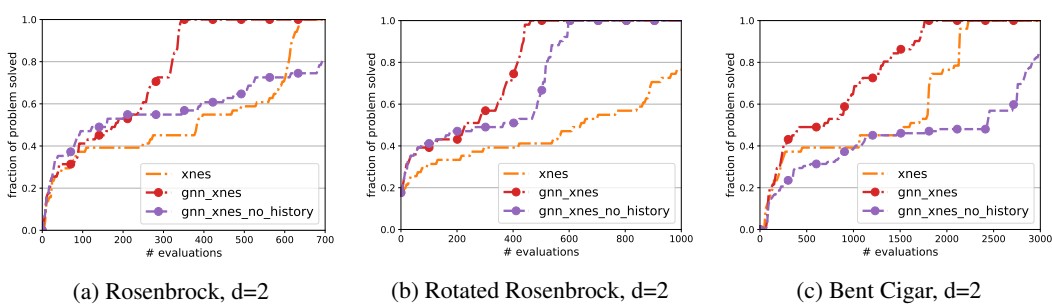

(a) Rosenbrock, d=2          (b) Rotated Rosenbrock, d=2          (c) Bent Cigar, d=2

Figure 13: ECDFs curves for xNES, GNN-xNES and GNN-xNES-no-history, for which the history size $T = 1$. Using past populations to estimate expectations improves the optimization.

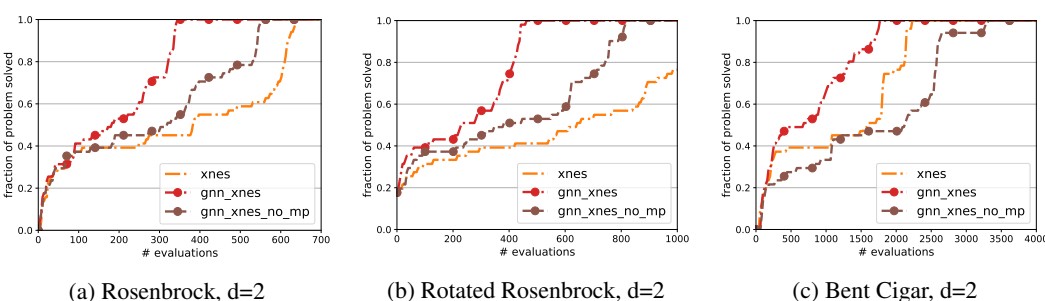

(a) Rosenbrock, d=2          (b) Rotated Rosenbrock, d=2          (c) Bent Cigar, d=2

Figure 14: ECDFs curves for xNES, GNN-xNES and GNN-xNES-nmp, which is not *mode preserving*. Ensuring that the training of the GNN doesn't impact the mode of the search distribution improves the optimization.

