# OpenReview forum: "Improving Evolutionary Strategies with Generative Neural Networks"
_ICLR.cc/2020/Conference — Reject_

### Official Review · AnonReviewer2 · 2019-10-21
**Official Blind Review #2**

**Rating:** 8

**Review:**

Review of “Improving Evolutionary Strategies with Generative Neural Networks”

Typically in ES, the distribution of solution candidates come from a hand-engineered distribution (i.e. multivariate Gaussian, or other parametrized distributions). In place of hand-engineered distribution choices, they introduce the use of GANs as a tool for Evolution Strategies. I liked the novelty of combining the use of GANs in new directions (namely ES/GA).

The core idea is to model the density function using Generative Neural Networks (GNN, MacKay 1995), and find the parameters of this GNN using tools from the normalizing flows literature for their NICE invertible properties (okay pun intended :) along with GAN-style training using historical data from the ES process.

They demonstrate their method on traditional blackbox optimization toy tasks (such as Rosenbrok and Rastrigin functions), and also on a few continuous control RL benchmark tasks, to demonstrate improved performance over a strong representative ES algorithm (XNES).

Overall, I liked the work as it provides a fresh way of using GANs with another subfield (ES/GA). If they want to improve the work, I would suggest demonstrating that their approach can solve certain difficult tasks that traditional ES methods (or RL methods) cannot solve. Although the experiments chosen are not difficult ones, I believe they were chosen for clarity to showcase the method, so I think that is fine (in case there are complaints that they experiments are too simple).

(For the record, I was looking to give a score of 7, but the ICLR system made me choose between 6 and 8, and I chose 8.)

**Experience Assessment:**

I have published one or two papers in this area.

**Review Assessment: Checking Correctness Of Derivations And Theory:**

I did not assess the derivations or theory.

**Review Assessment: Checking Correctness Of Experiments:**

I assessed the sensibility of the experiments.

**Review Assessment: Thoroughness In Paper Reading:**

I made a quick assessment of this paper.

---

> ### Author Response · Authors · 2019-11-09
> **Response to Reviewer2**
>
> We thank the reviewer for his supportive feedback.
>
> About the experiments: The experiments were indeed chosen to highlight situations were flexibility improves the ES procedure. Even if the synthethic examples look easy, they capture prototypical difficulties in (zeroth-order) optimization, such as high conditioning, high number of modes, asymetry, non-separability.. We can therefore expect that the good properties of GNN-ES that we highlighted in these experiments would extend to more general, "real-life" problems (this is what the BBOB testbed is designed for).
>
> We agree that future work could focus on applying our ideas to more difficult tasks, such as;
> 	- high dimensional tasks (for instance like the RL case we consider but with deep neural network policies)
> 	- conditional-task: this would be for instance the case with MDP-based RL. The challenge in this case is that the policy is usually stochastic, hence we would need to generalize our method for rewards corrupted with (potentially large) noise. Also, the application of this paper to MDP-based RL is not straightforward as one would need to generalize methods like PG, TRPO, PPO .. for our approach.
> 	- another possibly interesting line of application is the optimization of functions that are relying on small-dimensional manifold. Is this manifold is not "flat", we can expect our method to bring a clear improvement compared to classical ES (e.g with Gaussians).

---

### Official Review · AnonReviewer3 · 2019-10-21
**Official Blind Review #3**

**Rating:** 6

**Review:**

Summary: In ES the goal is to find a distribution pi_theta(x) such that the expected value of f(x) under this distribution is high. This can be optimized with REINFORCE or with more sophisticated methods based on the natural gradient. The functional form of pi_theta is almost always a Gaussian, but this isn't sufficiently flexible (e.g. multi-modal) to provide a good optimization algorithm. In response, the authors advocate for using a flexible family of generative neural networks for pi_theta. Using NICE as a generative model is desirable because it maintains volumes. This means that we can adjust volumes in latent space and this directly corresponds to volumes in x space. Doing so is useful to be able to tune how concentrated the search distribution is and to explicitly reason about the mode of the search distribution.

Overall, I found that there were a number of technical details that were well motivated, such as how to leverage the 'mode preservation' of NICE, how to use importance sampling to be able to use samples from multiple rounds of optimization when updating theta and the fact that any existing ES algorithm can be used to do the optimization in the latent space.



"We found that the PGES algorithm (naive stochastic gradient descent of (8) with the score-function estimator) applied to the NICE distribution suffers from the same limitations as when applied to the Gaussian; it is inable to precisely locate any local minimum."
   I don't understand this. Can't the Gaussian become very concentrated?

You write: "ES implicitly balance the need for exploration and exploitation of the optimization landscape. The exploitation phase consists in updating the search distribution, and exploration happens when samples are drawn from the search distribution’s tails."
This is a weak form of exploration, since there is no explicit mechanism that encourages f(x) to be evaluated at regions that it has never been evaluated on before. The search distribution's tails will have low probability mass, so exploration unlikely. Your proposed method uses a pi(x) that is flexible enough to represent multi-modal distributions. However, how can you ensure that your search procedure actually uses this flexibility? In other words, how is your proposed method any better at exploration that the baseline ES method?

It would be great to have a slightly more detailed alg. box in the main text for your proposed method, instead of having it in the appendix. Some details I found confusing, such as whether you perform one step of alternating optimization per call to f(x) or if you perform many steps of alternating optimization.

Your "Mode preserving properties" trick is cool. However, I don't fully understand how it is used. Surely you need to be able to be able to change the mode of the distribution some time? Do you only use the mode preservation trick at certain optimization steps? Again, incorporating this in the alg box would be helpful.

In the results, I was disappointed that you required restart strategies. I thought that one of the key advantages of using NICE was that you could capture a multi-modal search distribution. Can you explain?

 What do you mean by the 'global volume of the distribution?' What is the volume of a distribution? I understand what concept you're trying to convey, but can you be more precise?


**Experience Assessment:**

I have published in this field for several years.

**Review Assessment: Checking Correctness Of Derivations And Theory:**

I assessed the sensibility of the derivations and theory.

**Review Assessment: Checking Correctness Of Experiments:**

I assessed the sensibility of the experiments.

**Review Assessment: Thoroughness In Paper Reading:**

I read the paper thoroughly.

---

> ### Author Response · Authors · 2019-11-09
> **Response to Reviewer3**
>
> We thank the reviewer for his constructive feedback. We answer the reviewer's remarks below, in order.
>
> On the PGES algorithm: as we mention in Section 2.1 and as is documented in [1, Section 2.2], the PGES algorithm becomes highly unstable when a location-scale search distribution is concentrating, and is therefore unable to locate precisely a local minimum (this is one of the main reason behind the development of NES, i.e to be able to make the search distribution concentrate on a local minimum). What we report in the beginning of Section 4.1 is that, quite unsuprisingly, training a NICE search distribution with PGES suffers from the same limitations, which calls for more advanced training strategies.
>
> On the exploration/exploitation trade-off: the exploration mechanism in ES algorithms is indeed implicit, and is likely to succeed if the tails of the search distributions are well aligned with the level sets of the search distribution. When a search distribution is not adapted to the optimization landscape, it can either: a) concentrate but reduce its ability to explore or b) keep some entropy and therefore some exploration capacity but waste samples on regions that are known to be sub-optimal. What typically happens for ES algorithms with "standard" distributions is a), as pointed out by Figure 2. Flexible search distributions have greater chance of observing positive exploration samples, as they can shape their tails to adapt to the objective's geometry (i.e avoid sub-optimal regions while keeping some entropy). Figure 3 and 4 are good clues that the algorithm uses this flexibility to locally "fit" the objective landscape.
> Also, we believe that the fact that we are able to improve the performance of xNES with GNN-xNES using volume and mode-preserving transformations is a good clue that the exploration was better carried out with the GNN distribution - we explicitely enforced the GNN to have influence mostly on the tails of the search-distribution.
>
> Pseudo-algorithm: The pseudo-code for the generic algorithm had to be push to the supplementary materials because of space constraint. As adviced by the reviewer, we added details to the procedure to make it clearer.
> Alternating minimization: at each iteration of the algorithm, a population of \lambda points is sampled. The latent distribution is optimized thanks to the "base" ES algorithm (1 iteration), and then the GNN-part is optimized via off-policy training (several iterations of gradient-based optimization).
>
> Mode preserving: The mode-preserving trick is here to ensure that the GNN cannot change the mode of the distribution and focus on shaping its tails. The mode of the distribution can therefore change only if the mode of the latent distribution changes, which happens every time we apply the "base" ES algorithm to the latent distribution. As adviced by the reviewer, we added details on this procedure in the pseudo-algorithm section. We also modified the section dedicated to the mode-preserving property in an attempt to make it clearer.
>
> Restarts: Even if GNN-based distributions are more flexible than Gaussians, the multimodal objectives we consider have an extremely high number of modes and GNN-xNES can also miss the global minimum (we claim that this happens much less frequently with GNN-xNES than with xNES). We added a experimental results in the supplementary material to support that claim, showing that the numbers of restarts needed to find the global minimum is greater for xNES than GNN-xNES.
> Nevertheless, using restarts is necessary to provide a fair comparison between xNES and GNN-xNES; indeed, using restarts allows to compair algorithms that have different convergence rates; one algorithm can be very bad at detecting the global minimum but very fast at converging to a local minimum, while another converges very slowly to the global minimum. Restarting the first algorithm allows to fairly compare both algorithms (for further reference, this point is also discussed in [2]).
>
> Global volume of the distribution: as pointed out by the reviewer "global volume of the distribution" is not a well defined quantity (it can maybe be defined in 1d through quantiles, but we couldn't find a good definition in higher dimension). A better statement is that because the GNN forward-map has unit Jacobian, every measurable subset of the latent space that is assigned a given probability mass is mapped to a subset of the data space with the same probability mass. We modified the manuscript to make the concerned sentence more precise.
>
> [1] : Wierstra, Daan, et al. "Natural evolution strategies." The Journal of Machine Learning Research 15.1 (2014): 949-980.
> [2]: COCO: Performance Assessment, Hansen & al, https://arxiv.org/abs/1605.03560

---

### Official Review · AnonReviewer1 · 2019-10-23
**Official Blind Review #1**

**Rating:** 6

**Review:**

Summary:

As the title of the paper states, this paper tries to improve evolution strategies (ES) using a generative neural network. In the standard ES candidate solution is generated from a multivariate normal distribution, where the parameters of the distribution are adapted during the optimization process. The authors claim that the gaussian distribution, i.e., the ellipsoidal shape of the sampling distribution, is not adequate for the objective functions such as multimodal functions or functions with curved ridge levelsets such as the well-known Rosenbrock functions. The motivation is clearly stated. The technique is interesting and non-trivial. However, the experimental results are not very convincing to conclude that the proposed approach achieves the stated goal. Moreover, this paper may fit more to optimization conferences such as GECCO.

Because of the empirical results, I would rate this paper as the border line (around 5), but due to the slightly annoying rating system the rate appears as 6.

Comments:

P2: "Efficient Natural Evolutionary Strategies (xNES) (Sun et al., 2009) has been shown to reach state- of-the-art performances on a large ES benchmark."

This algorithm is "eNES" and  this algorithm is not competitive with the state-of-the-art ES such as CMA-ES. The authors might want to refer to exponential NES, which is xNES, proposed by Glasmachers et al 2010.

P5: "Indeed, other bijective GNN models like the Real-NVP (Dinh et al., 2016) introduce non-volume preserving transformations, which can easily overfit and lead to premature concentration and convergence." Has it been reported in a reference? If so provide the reference. If not, the authors should state that it has been observed the authors preliminary study. In any case, I think it depends how the model is used or trained, and this statement itself is not universally true.

P7: "By using fη instead of gη as the push-forward map of the NICE model, we ensure that the flexibility brought by the GNN only impacts the tails of the search distribution. As detailed in an ablation study presented in Appendix F, this additional tool turns out to be essential in order to use GNNs for ES."

I barely understood this point. Please make is clearer.

P8: Experimental results are not very convincing. The experiments are limited to dimension 2, 5, 10 and only a few functions are selected from the BBOB test function suite. How about on 20D? What happens if the target is 1e-8, which is the default setting in BBOB?

Figure 3 looks interesting, and this is what the authors are trying to achieve. Therefore, it looks like the authors reached the stated objective. However, this is only 2D. No results are provided to convince that the proposed strategy achieved the stated objective.

Figure 4  simply looks that the proposed algorithm failed to reach the "flexibility" stated in Section 2: "Another limitation of classical search distribution is their inability to follow multiple hypothesis, that is to explore at the same time different local minima. Even if mixture models can show such flexibility, hyper-parameters like the number of mixtures have optimal values that are impossible to guess a priori."

From these results, I am not convinced that the proposed strategy really achieved more flexible distribution than the classical methods, and whether the flexibility contributes to improve the performance.

Another critical point to be discussed is its usefulness. Since this algorithm is proposed to "improve evolution strategy" as a black-box optimizer (not for specific tasks), I expect to improve the state-of-the-art performance. Are the reported results outperform the CMA-ES? Based on Glasmachers et al (2010), xNES tends to require more objective function evaluations than CMA-ES, especially for higher dimensional cases. I am curious to know if the proposed approach outperforms the CMA-ES on Rosenbrock functions.



**Experience Assessment:**

I have published in this field for several years.

**Review Assessment: Checking Correctness Of Derivations And Theory:**

I carefully checked the derivations and theory.

**Review Assessment: Checking Correctness Of Experiments:**

I carefully checked the experiments.

**Review Assessment: Thoroughness In Paper Reading:**

I read the paper thoroughly.

---

> ### Author Response · Authors · 2019-11-09
> **Response to Reviewer1**
>
> First, we wish to thank the reviewer for his constructive remarks and comments.
>
> Wrong citation: as the reviewer pointed out, there was a mix-up in the names and references when introducing exponential NES (xNES), and we cited the wrong paper. This has now been fixed in the manuscript.
>
> Remark on Real-NVP: the comment made about the premature convergence of ES augmented with Real-NVP is based on our empirical study. We clarified this point in the manuscript. For all the training methods we used, we noticed that Real-NVP-based ESs often suffered from premature convergence or were much slower than NICE-based-ESs because of their entropy decreasing faster. Also, on a more technical aspect, NICE revealed to be much more stable numerically than Real-NVP when the distribution is concentrating.
>
> About the mode preserving mechanism: one of the goal of our paper was to study how flexibility (brought by GNN) could improve the exploration in ES. As discussed during Section 2.2, exploration in ES is implicit and happens when samples are drawn from the tails of the search distribution. We therefore want the optimization of the GNN to focus on aligning its tails with the level sets of the objective function, and not to simply "move around" the search distribution. Our mode preserving trick allows this by enforcing that the mode of the search distribution cannot change when the GNN is trained. We modified the manuscript to make this point clearer.
>
> On the experimental results: we believe that the results we report in the main text are good indicators that our algorithm works well and improves xNES. For unimodal functions, the average speed-up is often greater than several hundreds function evaluations (which with the considered populations sizes corresponds to dozens of algorithm iterations). For multimodal functions, the speed-up can be quite significant (dimension 10 on Rastrigin and Griewank-Rosenbrock for instance; notice that for visualization purposes the number of evaluations to reach the target is scaled by the square of the dimension). On the choices of functions; for unimodal functions, the choice from BBOB is rather limited as almost all functions have ellipsoid level sets, where the Gaussian dsitribution is already perfectly adapted. Within the main text and the supplementary material, we presented results on all unimodal functions that have non-ellipsoidal level sets in the BBOB testbed.
> Regarding the target set to 10^{-5}; at this point, the local minimum is well located by the algorithm, which only needs to exploit. Under this regime, GNN-xNES no longer has impact and its advantage remains the same. We set the target to 10^{-5} for visualization purposes, in order to better visualize the difference between xNES and GNN-xNES in the early stage of optimization.
> As for density visualization, we believe that Figure 3,4 and Figure 8 in Appendix are good examples that GNN-based search distribution indeed leverage the flexibility of the GNN to "fit" the level curve of the objective function and follow several minimum in a more efficient way than the Gaussian. We are not sure how to provide similar plots in higher dimensions; displaying projections of the objective function and of the search distribution density on planes can be misleading (even a unimodal density can look multimodal when projected on a plane).
>
> On usefullness: our choice to focus on NES in the experimental section is motivated by the interest in that family of algorithms in the machine learning community (for instance in non-MDP-based RL). We showed that GNN-xNES improves xNES. However, this experimental choice is rather arbitrary, as we have showed that our GNN-based method can act as a plugin for any ES algorithms (henceforth including CMA-ES). Therefore, one can easily implement a GNN-based CMA-ES extension in the same fashion as for xNES. We added figures in the supplementary material showing that this extension improves CMA-ES on the Rosenbrock function.

---

### Decision · Program_Chairs · 2019-12-19

**Decision:**

Reject

**Comment:**

Evolutionary strategies are a popular class of method for black-box gradient-free optimization and involve iteratively fitting a distribution from which to sample promising input candidates to evaluate.  CMA-ES involves fitting a Gaussian distribution and has achieved state-of-the-art performance on a variety of black-box optimization benchmarks when the underlying function is cheap to evaluate.  In this work the authors replace this distribution instead with a much more flexible deep generative model (i.e. NICE). They demonstrate empirically that this method is effective on a number of synthetic global optimization benchmarks (e.g. Rosenbrock) and three direct policy search reinforcement learning problems.  The reviewers all believe the paper is above borderline for acceptance.  However, two of the reviewers said they were on the low end of their respective scores (i.e. one wanted to give a 5 instead of a 6 and another a 7 instead of 8.)  A major issue among the reviewers was the experiments, which they noted were simple and not very convincing (with one reviewer disagreeing).  The synthetic global optimization problems do seem somewhat simple.  In the RL problems, it's not obvious that the proposed method is statistically significantly better, i.e. the error bars are overlapping considerably.   Thus the recommendation is to reject.  Hopefully stronger experiments and incorporating the reviewer comments in the manuscript will make this a stronger paper for a future conference.